# It Is What the Surgeon Does Not See That Kills the Patient

**DOI:** 10.3390/jcm13082238

**Published:** 2024-04-12

**Authors:** Paul H. Sugarbaker

**Affiliations:** Program in Peritoneal Surface Malignancy, Washington Cancer Institute, Washington, DC 20007, USA; paul.sugarbaker@outlook.com

**Keywords:** colon cancer, CT, CEA, CA19-9, CA-125, colonoscopy, neoadjuvant chemotherapy, cytoreductive surgery, peritonectomy

## Abstract

Background: Patients with colon cancer may present at multiple different stages of the disease process. Many patients can be cured of colon cancer as a result of a simple surgical procedure usually performed by minimally invasive techniques. However, there are a variable number of patients, estimated at approximately 10%, who have a more advanced disease. If these patients are treated by the current conventional standard of care, the likelihood for treatment failure is extremely high. Methods: These are not patients with known disseminated disease but patients who are at high risk of recurrent disease unless special treatments are initiated preoperatively and intraoperatively. The identification of these patients is by (1) a high-quality CT scan, (2) tumor markers found preoperatively, (3) colonoscopic findings, and (4) symptoms. Results: Patients identified as being at high risk require special preoperative treatments which include neoadjuvant chemotherapy. Intraoperative chemotherapy with HIPEC should occur as part of the treatment if peritoneal metastases are documented by biopsy. In the operating room, a thorough exploration of all possible occult peritoneal spaces for metastatic disease needs to be performed. A modified cytoreductive surgical procedure along with a colon resection is performed in order to minimize sites of occult peritoneal metastases. This includes the greater omentum, ovaries, and tubes in postmenopausal women. Peritonectomy is used to create a shroud around the tumor so that all peritoneum that has been in direct contact with the tumor surface is resected and is used as a barrier against tumor cell dissemination in the process of colon cancer resection. If peritoneal metastases are visualized at any site, HIPEC should be included as part of the treatment package. Conclusions: I am convinced that patients at high risk of recurrence will have an improved outcome with proper preoperative evaluation, preoperative neoadjuvant chemotherapy, and a revised intraoperative management strategy.

## 1. Introduction

### 1.1. Case Presentation

27 November 2017: A 59-year-old otherwise healthy woman underwent a screening colonoscopy. Biopsy from the right colon showed poorly differentiated adenocarcinoma with near circumferential involvement of the lumen of the ascending colon (CEA = 90 ng/mL).

13 December 2017: Preoperative CT showed an 8 cm mass (Figure 1).

16 December 2017: A laparoscopic right colon resection was attempted. The surgical approach was converted to open because of possible injury to the right ureter. The surgeon classified the resection as R0. 

21 December 2017: Pathology report confirmed a poorly differentiated (grade 3) tumor with negative margins. pTNM was T3N2Mx with 7 of 35 positive lymph nodes. Adjuvant chemotherapy with FOLFOX was initiated. 

27 May 2018: Patient was asymptomatic. CEA increased from 3.0 to 20 ng/mL. CT showed a mass in the left paracolic gutter. Biopsy of this mass showed adenocarcinoma. A 1.5 cm mass was present at the ileocolic anastomotic site (Figure 2). A second mass was demonstrated in the rectovesical space (Figure 3). FOLFIRI was initiated. CEA increased to 74 ng/mL.

22 May 2018: Patient evaluated for cytoreductive surgery (CRS) plus hyperthermic intraperitoneal chemotherapy (HIPEC) with melphalan 20 mg/m^2^. 

Preoperative peritoneal cancer index PCI = 7. 

Laparoscopy was not performed. 

### 1.2. Procedures

Total resection of old abdominal incision using skin traction sutures. 

Parietal peritonectomy abdominopelvic regions 0, 4–8. 

Repeat resection of old ileocolic anastomotic site with re-anastomosis. 

Rectosigmoid colon resection with two-layer descending colon to rectal anastomosis. 

Time for CRS plus HIPEC by open method—8 h, PCI = 16.

Completeness of cytoreduction (CC score) = 0. 

No grade 3 adverse events. Length of stay = 18 days. 

Status = Dead of disease 4 years after CRS. Subsequent systemic chemotherapy treatments unknown. 

### 1.3. Case Discussion

What could have been done differently with this patient? The 4-year survival of a patient who developed peritoneal metastases may be acceptable to some oncologists. Others, including myself, will insist that major changes in the management of the primary right colon cancer could have resulted in a cure or a more prolonged survival of this patient. Some oncologists will interpret this patient’s death from cancer progression as a result of an aggressive tumor biology that could not be controlled by standard of care management. Others will insist that more knowledgeable preoperative and intraoperative management may have mitigated surgical trauma to the resected cancer specimen and the local–regional and distant dissemination that caused recurrent disease. It may have been possible to avoid a dissemination of cancer cells within the abdomen and pelvis that caused treatment failure in this patient. 

Not all primary colon cancers are in an advanced stage, as in this 59-year-old asymptomatic patient. My hypothesis is that 90% of our treatment failures with colon cancer occur in 10% of the patients. The preoperative identification of these high-risk patients is a crucial step in improving the outcome with an advanced disease. The second step is the initiation of management plans preoperatively and intraoperatively that perfect the clearance of the primary colon cancer and its containment prior to and at the time of resection. It is completely possible that this 59-year-old patient entered the operating room with a contained primary cancer but left the operating theater with disseminated disease as a result of cancer spread from surgical trauma related to the cancer resection.

The first line of defense against peritoneal metastases of the surfaces of the abdomen and pelvis is an intact peritoneal surface. The efficiency of implantation of cancer cells that the surgeon cannot see is greatly increased at sites of inflammation and traumatized peritoneum [1]. Cancer cells free in the peritoneal space or recently deposited on peritoneal surfaces as a result of surgical trauma must be prevented. Cancer cells caught up in an inflammatory exudate, fibrin, or blood clot are likely to progress. For primary colon cancer, complete clearance and complete containment must be the goals of an optimal resection. 

### 1.4. Clinical and Laboratory Signs That Should Be Used to Identify a Colon Cancer Patient at High Risk of Treatment Failure

The preoperative CT scan read by the surgeon and a knowledgeable radiologist is a crucial assessment of future treatment failure (Table 1). Many of the concerning radiologic features for a high-risk primary colon cancer were displayed by the patient. The identification of concerning CT features that accurately predict outcome (as in patients with peritoneal mesothelioma) has not yet been achieved [2,3]. A list of concerning CT features for primary colon cancer correlated with disease-free survival and overall survival does not exist. However, CT interpretation, one patient at a time, and consolidated with the clinical and laboratory data presented in the following paragraphs of this manuscript, will allow for the identification of a majority of patients at high risk of treatment failure. Although CT was a standalone prognosticator in the FOXTROT randomized trial, the CT must be interpreted along with all other prognostic indicators [4]. In a composite evaluation, CT may indicate that the patient is at high risk of treatment failure and needs a specialized treatment plan. Several attempts by radiologists using CT to stage colon cancer patients preoperatively have been published. Nerad and coworkers published a systematic review and meta-analysis regarding the accuracy of CT for the local staging of colon cancer [5]. In their evaluation of 13 manuscripts, they found that CT had a sensitivity of 90% and a specificity of 69% to detect T3 or T4 tumors. Sensitivity for lymph nodal involvement was 78% and specificity was 68%. Nerad and coworkers evaluated the CT findings according to pathology reports. Data regarding the incidence of peritoneal metastases associated with T-stage or N-stage were not available. Also, other poor prognostic findings from the CT such as greatest tumor diameter, involvement of adjacent organs, and absence of fat planes between the cancer and vital structures. One can conclude that preoperative CT must be an important part of the preoperative evaluation of patients with colon cancer. It is useful along with other clinical, laboratory, and histologic findings to identify patients at high risk of treatment failure.

The suggestions of a world expert in the radiologic assessment of primary colon cancer may be appropriate in an attempt to establish the current state of our knowledge [6]. Brown asks that the CT be performed in a clean colon with oral contrast distributed throughout the bowel. Although small tumors may be poorly characterized by CT, more advanced and aggressive cancers provide the radiologist with useful information regarding outcome. Other sites of disease outside of the colon in the liver, lung, and peritoneal cavity are accurately detected. Bowel wall invasion with nodules extending 5 mm or more beyond the muscularis propria is an indication of a T3 or T4 radiologic stage and an increased incidence of local recurrence and peritoneal dissemination. As shown in Table 1, other poor prognosis findings include lymph node or vascular invasion. Also, encroachment upon organs or structures in contact with the primary disease in the absence of a fat plane between them suggests penetration through the bowel wall with invasion of the adjacent tissue. The size of the primary cancer with a tumor diameter greater than 5 cm is in itself an indicator of advanced disease. These radiologic findings taken together indicate a radiologic TNM that can establish the colon cancer patient in the group at high risk of surgical treatment failure. If the CT does not provide definitive information, a preoperative MRI is indicated. The MRI is expected to be especially useful with mucinous cancers [3].

The tumor markers carcinoembryonic antigen (CEA), cancer antigen 19-9 (CA 19-9), and cancer antigen 125 (CA-125) can be used to estimate the extent of disease in patients with primary colon cancer. A CEA greater than 5 ng/mL indicates reduced survival [7]. Also, the higher the CEA, the greater the reduction in survival. Simultaneously positive preoperative CEA, CA 19-9, and CA-125 had the highest rate for death and shortest survival time [8]. Preoperative colon cancer patients should have all three tumor markers CEA, CA 19-9, and CA-125 assessed within a few days prior to the initiation of treatment. Tumor markers help to identify colon cancer patients preoperatively as having a high risk of surgical treatment failure.

Colonoscopic biopsy and architecture of the primary cancer have prognostic implications. If the biopsy shows an undifferentiated histology, as in the patient presented, the prognosis is guarded regarding long-term survival [9]. Other poor prognosis histologies would be signet ring cell carcinoma, perineural invasion, intramural vascular invasion, lymphatic invasion, and involvement of the circumferential margin. If sufficient tissue is available, the genomic classification as hypermutated (high microsatellite instability) versus non-hypermutated is of great benefit in planning neoadjuvant treatments. Cancers that are microsatellite-unstable are candidates for anti-PDL1 therapy [7]. Cancers that do not show chromosomal instability are candidates for neoadjuvant chemotherapy [4].

Symptomatic patients are at higher risk of death from their cancer diagnosis [10]. Pain, a palpable mass, weight loss, symptoms of obstruction, and perforation are all associated with early treatment failure. A symptomatic patient deserves special concern for early treatment failure and an adverse outcome. 

### 1.5. Individualized Evaluation, Neoadjuvant Chemotherapy, and Extended Surgery Recommended by the MDT for Colon Cancer Patients

In the FOXTROT trial, the sole indicator of advanced disease was the T3 or T4 bowel wall thickening [4]. Improved disease-free survival (HR 0.72, *p* = 0.037) was present when these patients with advanced disease were treated with FOLFOX neoadjuvant chemotherapy (NAC). However, the CT assessment used by the FOXTROT trial lacked both sensitivity and specificity for the detection of patients at high risk of treatment failure. Tumor regression was correlated strongly with freedom from recurrence. However, the number of patients misclassified radiologically that were treated as being high-risk but were not pathologically T3/T4 and the number of patients thought to be radiologically at high risk but would have survived with conventional treatment are not known. The accuracy of abdominal CT as used in the FOXTROT randomized trial to detect those at high risk of recurrence of colon cancer is not accurate, nor is pelvic MRI to detect advanced rectal cancer. Ongoing advances in radiologic techniques should help to identify patients who may profit from NAC compared to patients who should have conventional surgery. 

### 1.6. High Risk Determined by a Composite Analysis of Radiologic and Clinical Features

I used a composite analysis of radiologic features and clinical features to improve the accuracy of diagnosis. Sugarbaker et al. showed that a composite evaluation of CEA and liver scan markedly increased the specificity of diagnosis of metastatic breast cancer in the liver [11]. Similarly, the MDT should focus on an individual patient who has radiologic signs that suggest a high likelihood of treatment failure. When a question regarding high risk is identified by preoperative CT, other manifestations of high-risk disease should be sought. If the extent of high risk reaches or exceeds 100%, as shown in Table 1, the patient should be recommended for NAC. One exception to this recommendation is patients who show high microsatellite instability. These hypermutated patients respond poorly or not at all to systemic chemotherapy and would not profit from NAC. 

Perhaps the most difficult decision that must be made regarding these colon cancer patients concerns which patients are placed in the group at high risk of treatment failure. This should be a unanimous decision of the Multidisciplinary Team (MDT). As with any important and far reaching treatment-related decision, all of the information listed in Table 1 should be available for all participants in the MDT to consider. The MDT must accept the fact that no mathematical equation exists at this point in time determining whether to treat a patient at high risk of treatment failure versus using treatment as a standard of care. There is a small possibility that a patient will be overtreated, in that neoadjuvant chemotherapy may not be necessary and the modified CRS may be unnecessary. 

My suggestion is that the radiologist with all other members of the MDT review a recent CT (less than 2 weeks old). Of course, disease outside of the abdomen and pelvis will disqualify the patient. The concerning CT features that maintain the patient as a possible candidate for the high-risk group would include T3/T4 bowel wall invasion, suspicion of positive lymph nodes, and encroachment of the mass on surrounding structures. Then, the tumor markers CEA, CA19-9, and CA-125 should be reviewed. Any tumor marker greater than three times the upper limit of normal (CEA ≥ 15 ng/mL, CA19-9 ≥ 120 u/mL, and CA-125 ≥ 105 u/mL) suggests that the patient should be considered in the high-risk group. These values are somewhat arbitrary. More than one tumor marker in the abnormal range, even though the levels would not be three times the upper limit of normal, suggests a high risk [8]. A colonoscopic biopsy that shows aggressive histology maintains the patient in the high-risk group. This includes poorly differentiated histology, signet ring cells, and lymphovascular or perineural invasion. Circumferential involvement of the colon wall indicates high risk. Finally, if the patient is symptomatic, he/she is more likely to be at high risk of surgical treatment failure. 

A reasonable judgement regarding the placement of a patient in a high-risk group would include a score of 100% or more from the list presented in Table 1.

### 1.7. Preoperative Management Strategies for Patients at High Risk of Local–Regional Treatment Failure

Patients with colon cancer who are at high risk of local recurrence and peritoneal metastases as a result of cancer cells disseminated within the abdomen and pelvis must have preoperative treatments initiated prior to a conventional resection. The percent of high-risk patients will vary with the quality of the healthcare system and the socioeconomic status of the patient population. If the patient is identified as being at high risk of treatment failure, as seen in Table 1, the colon resection must be delayed. If the patient is not hypermutated by colonoscopic biopsy, the patient should receive neoadjuvant chemotherapy as recommended in the FOXTROT trial [4]. Oxaliplatin and 5-fluorouracil were given using a modified FOLFOX schedule, as follows: oxaliplatin 85 mg/m^2^ plus l-folinic acid 175 mg for a 2 h infusion, fluorouracil 400 mg/m^2^ bolus, and fluorouracil 2400 mg/m^2^ for a 46 h infusion, repeated once every 2 weeks. A total of 12 weeks of treatment was planned. Dose reductions, treatment delays, and early cessation for toxicity were permissible, as in routine practice. 

Prior to surgery as part of the consent for surgery, permission to use HIPEC is to be obtained. If any peritoneal metastases are documented, HIPEC should be given as a planned part of this resection [12]. If the tumor must be peeled away from vital structures such as the pancreas, aorta, iliac vessels, and major veins such as the vena cava, HIPEC should be given. This HIPEC treatment is an attempt to eradicate cancer cells spilled as part of an R-1 resection. 

### 1.8. Intraoperative Treatment Strategies for Patients at High Risk of Local–Regional Treatment Failure

The surgery should proceed as a modified CRS in order to remove any anatomic sites that may contain cancer cells [13]. A greater omentectomy is indicated. Resection of the ovaries and fallopian tubes is indicated if the patient is postmenopausal. The round ligament and falciform ligament are resected and the hepatic bridge is opened. All parietal peritoneum that was in direct contact with the primary tumor is removed by peritonectomy. All peritoneal spaces known to harbor occult peritoneal metastases are visualized, irrigated, and suctioned. The colon resection is performed with wide margins and a lymph node dissection to the superior mesenteric vein on the right and to the origin of the inferior mesenteric vein on the left. Extensive saline washing removes all clotted blood that may have accumulated (Table 2). All specimens are submitted separately to the pathologist. All cancer specimens are assessed for tumor regression grade [7]. If any evidence of visible peritoneal metastases are documented, HIPEC with mitomycin C is administered. 

Colon cancer resection is a highly invasive surgical procedure that requires the removal of 20 to 40 cm of the large intestine and an anastomosis of the proximal and distal ends of the bowel. The resection may be complicated by surgical site infection and/or disruption of the anastomosis. Multiple other adverse events have been described. Nevertheless, surgical removal of the primary colon cancer must occur because it is the only curative treatment available today. Recently, a “Bundled infection reduction program” was published by Badia et al. [14]. In this multicenter study, a variety of antibiotic treatments and bowel cleansing treatments were implemented. This included systemic antibiotic prophylaxis, oral (intracolonic) antibiotic prophylaxis, mechanical bowel preparation, a laparoscopic colon resection, normothermia, and wound protection. All six treatments were bundled together for use in every patient. The overall surgical site infection fell from 18.38% to 10.17% (odds ratio: 0.503 [0.473–0.524]). The authors concluded that the implementation of this care bundle resulted in a significant reduction in surgical site infection. 

In some, but not all, patients identified as having advanced primary colon cancer, the portion of the bowel containing the cancer must be removed along with organs or structures adherent to the cancer [15]. This en-bloc resection is an attempt to not only clear the primary cancer but also contain it. These extended resections will reduce the spillage of cancer cells that may result in local recurrence or peritoneal metastases. These multivisceral resections result in an increased complication rate (47.2% vs. 43.7%, *p* < 0.001) and increased mortality rate (4.9% vs. 3.8%, *p* < 0.001). The 5-year overall survival rate in non-multivisceral resections was 69.5%, and in the group with multivisceral resections it was 53.9% (*p* < 0.001). Although survival was reduced in the high-risk group of patients requiring extended surgical procedures, it did occur. The initiation of innovative preoperative and intraoperative treatments is an attempt to improve the survival of patients with advanced disease. 

### 1.9. Consequences of Undertreating vs. Overtreating Colon Cancer Patients at High Risk of Local Recurrence and Peritoneal Metastases

The decision by the MDT to treat a patient using the standard of care versus the protocol for those at high risk of treatment failure will be an incorrect decision in some patients. The accuracy of the guidelines based on clinical and radiologic parameters is to be evaluated at the time of the pathology report. Some patients will be overtreated. Overtreatment has been a criticism of the FOXTROT randomized trial [4]. However, the consequences of overtreatment versus undertreatment are worthy of comparison. If a patient with colon cancer is not at high risk and is treated on the high risk protocol, they may receive four cycles of FOLFOX that they may not need. Also, they will have additional resections of non-vital structures. Morbidity and mortality may be increased. The length of hospital stay may be prolonged 2–3 days because of the peritonectomy. No long-term adverse events are expected. In contrast, if a high-risk patient is treated by conventional surgery and local recurrence and/or peritoneal metastases occurs, there will be severe long-term consequences. The likelihood of death from local–regional recurrence of cancer approaches 80%. Only approximately 25% of peritoneal metastases patients are salvaged by complete CRS plus HIPEC [16]. Having some patients be overtreated compared to undertreated is preferable. 

### 1.10. Progress in the Management of Colon Cancer, Similar to the Major Advances in the Local–Regional Control of Rectal Cancer

There can be no doubt that a major improvement in the local–regional control of rectal cancer has occurred in the last two decades. The incidence of local recurrence and peritoneal metastases in this disease has varied between 20 and 40% in multiple clinical studies of rectal cancer resection using surgery alone. As the use of neoadjuvant chemoradiation therapy and the resection of the total mesorectal envelope have become a standard of practice, local–regional recurrence of rectal cancer has decreased to 3–4%. In some patients, especially those with early rectal cancer, a watch-and-wait policy following treatment with chemoradiation therapy is used, with surgery being recommended only for those patients who have a progressive disease. The goal of our studies with colon cancer is to treat patients at high risk of treatment failure prior to a surgical intervention, as has occurred with rectal cancer. After preoperative NAC, a surgical intervention that will achieve complete containment of the colon cancer will be utilized. Through a careful preoperative evaluation of patients using radiology, tumor markers, colonoscopy, and symptoms/signs, only those who pass the threshold for a high risk of surgical treatment failure will be recommended to receive preoperative chemotherapy and intraoperative modified cytoreductive surgery. In summary, the advances made in the management of rectal cancer are now being explored for colon cancer.

### 1.11. Use of Deep Learning for Improved Selection of Patients at High Risk

As the continued use of the selection guidelines for the identification of primary colon cancer patients who are at high risk of recurrence occurs, improvements in this process may emerge. For preoperative CT, deep learning may be used to select images that indicate a colon cancer at high risk of surgical treatment failure. Radiologists, surgeons, and medical oncologists must first identify by consensus the concerning CT features that they associate with high risk. This would be similar to the methods used with peritoneal mesothelioma patients to predict the outcome of treatment of this disease [2]. Following this, the computer must be trained to identify these images, even going beyond the current accuracy of detection of high-risk cancers by the unaided eye. As copious data are accumulated from the images of high-risk cancers that have recurred, AI will increase the identification of patients who need special treatment strategies. Sahoo and coworkers used a deep learning approach to localize colon cancer lesions within the large bowel [17]. The sensitivity with AI localization was 0.83 (±0.29) and the specificity was 0.97 (±0.01). These investigators did not, as yet, attempt to characterize the colon cancers as high- or low-risk. Their report was, at this point in time, concerned only with localization. Wang et al. and Yuan et al. used an image-based deep learning algorithm to improve the detection of synchronous peritoneal metastases in patients with primary colon cancer. This information which is available preoperatively alerts the surgeon to the special need for cytoreductive surgery and HIPEC for selected patients [18,19].

A colonoscopic biopsy of cancer tissue that is representative of the histopathology of the primary tumor mass can predict the outcome of surgical treatment. Zhou et al. used deep learning models to interpret whole-slide images. A quantitative analysis showed that high cellularity, high ratios of tumor cells to background stroma, large tumor nuclei, and low immune infiltration are indicators of a high risk of surgical treatment failure [20]. Carcagni et al. used colon carcinoma grading by AI to improve the accuracy and grading of biopsies. Deep learning served to speed up the reading of the histopathology and improve accuracy and interobserver inconsistency [21]. Deep learning presents algorithms to reliably predict prognosis from a colonoscopic biopsy.

## Figures and Tables

**Figure 1 jcm-13-02238-f001:**
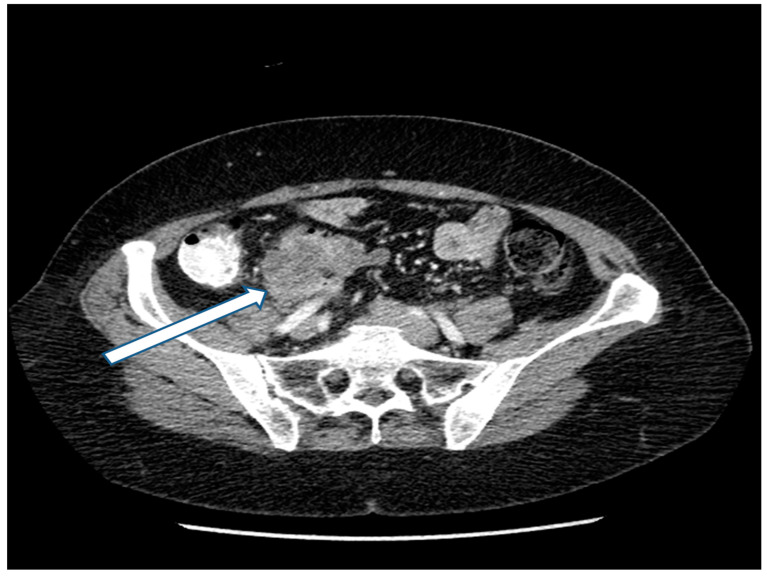
CT of a right-sided primary colonic mass that measures 8 cm in its greatest dimension. The external surface of the mass is irregular, suggesting peritoneal thickening. The medial aspect of the mass abuts the duodenum and no fat plane separating the mass from the second portion of the duodenum is evident. The mass penetrates posteriorly toward the right common iliac artery without obvious obstruction of the right ureter. A fat plane between the posterior aspect of the mass and the psoas muscle is present. No obstruction or perforation of the bowel above the mass exist. A single 1.5 cm lymph node is enlarged at the medial aspect of the mass. No additional systemic or local–regional manifestations of the cancer are evident. Radiologically, the mass is staged as T3/T4, N1, M0 with transmural extension through the wall of the intestine.

**Figure 2 jcm-13-02238-f002:**
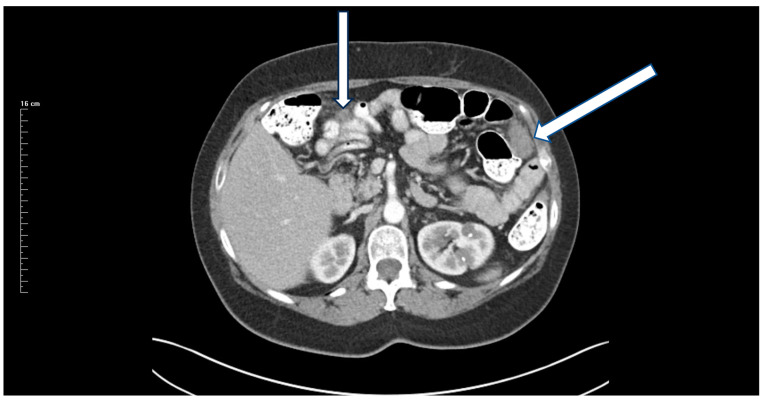
CT demonstration at the time of recurrence of a mass in the left paracolic sulcus. CT-guided biopsy showed adenocarcinoma.

**Figure 3 jcm-13-02238-f003:**
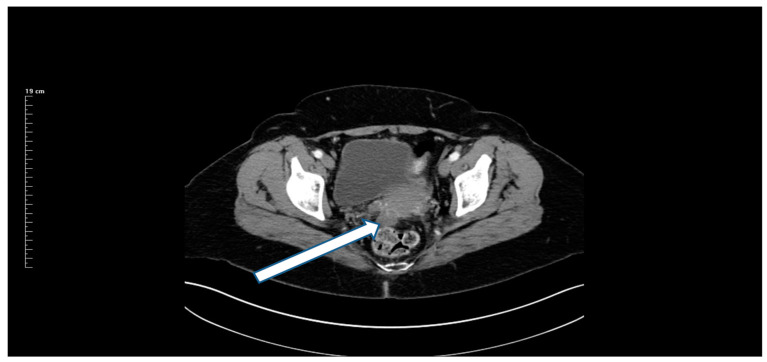
CT demonstration at the time of recurrence of a mass in the rectovesical space.

**Table 1 jcm-13-02238-t001:** Guidelines for selection by the MDT of patients for the high-risk group. Patients who reach 100% are recommended for inclusion in the high-risk group.

Evaluation	Score
** *Preoperative CT* **	
Bowel wall invasion more than 5 mm beyond muscularis propria (T3 or T4)	25%
Lymph node or vascular invasion	25%
Tumor diameter ≥5 cm or encroachment on adjacent organs or structures in the absence of a fat plane	25%
** *Tumor markers CEA, CA 19-9, or CA-125* **	
Any one of the three tumor markers >3 times the upper limit of normal	25%
More than one tumor marker in the abnormal range	25%
** *Colonoscopic findings* **	
Aggressive histology including poorly differentiated, signet ring cells, and lymphovascular or perineural invasion	25%
Circumferential involvement of the bowel wall	25%
** *Symptoms* **	
Symptoms present including a palpable mass, pain, obstruction, localized perforation, weight loss	25%

**Table 2 jcm-13-02238-t002:** Techniques required in a modified CRS with colon resection for patients scored as being at high risk of treatment failure.

Greater omentectomy Ovaries and fallopian tubes in postmenopausal women Falciform and round ligament Division of the hepatic bridge Peritonectomy of all parietal peritoneum in direct contact with the primary tumor Adjacent organs or structures invaded by the primary cancer are to be removed en bloc

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
