# Peer review of "It Is What the Surgeon Does Not See That Kills the Patient"

_jcm, 2024, doi:10.3390/jcm13082238_

Round 1

Reviewer 1 Report

Comments and Suggestions for Authors

I would like to congratulate the authors on their fascinating work regarding this interesting report entitled “It is what the surgeon does not see that kills the patient”. The manuscript is well-written and the incorporated figures make the study easy to follow.

1) I would like a brief discussion on postoperative complication after colorectal surgery

2) The SSI is the most common postoperative complication after colorectal surgery, causing pain and suffering to patients. In addition, this complication has been associated with negative economic impact, increased morbidity, extended postoperative hospital stay, readmission, sepsis, and death. I would suggest adding this important information to the discussion section and consider citing the recently published articles

https://pubmed.ncbi.nlm.nih.gov/34080408/

3) I would suggest a brief discussion on the role of deep learning in Colorectal Cancer (CRC) diagnosis. According to the literature, further investigation has been conducted into the potential clinical practice implementation of deep learning algorithms for the classification and diagnosis of CRC histopathology images. The advancement made possible by deep learning algorithms has the potential to improve CRC detection's accuracy and efficacy.

Author Response

Reviewer 1:

Comments 1 and 2 regarding postoperative complications and surgical site infection in the surgical management of colon cancer: A paragraph on adverse events associated with the surgical management of colon cancer was added to Section 1.7. Also, a reference and discussion of the advanced surgical procedures patients in the high risk group sometimes required was added. The more aggressive surgery sometimes required in a high risk colon cancer results in increased morbidity and mortality.

Comment 3 role of deep learning in colon cancer diagnosis: The 4 preoperative diagnostic tests recommended in the manuscript for selection of patients by the MDT for placement of patients in a high risk group include CT, tumor markers, colonoscopy and symptoms/signs. There is progress regarding the use of AI to read preoperative CT. Recent references to support this advance in the use of AI for interpreting the CT were added to the manuscript in Section 1.11.

No manuscripts regarding the use of AI with tumor markers in primary colon cancer are currently available. Support from the literature regarding the use of deep learning to improve the accuracy of grading of colon adenocarcinoma has been published. The possible application of this new approach of histopathology to assess the risk of surgical treatment failure was presented. Also, in addition to stratification of colonoscopic biopsy, the gross configuration of the primary cancer as visualized and photographed by the endoscopist had not been subjected to evaluation by AI. This is definitely possible. Symptoms and signs have not been evaluated by AI. As requested by Reviewer 1, paragraphs regarding future prospects for improved selection factors for identification of high risk groups using deep learning were added as final paragraphs of this manuscript.

Reviewer 2 Report

Comments and Suggestions for Authors

The authors showed interesting clinical observations: the higher the CEA, the greater the reduction in survival. At the same time, positive preoperative CEA, CA 19-9 and CA-125 results had the highest mortality rate and short-term survival estimates. Preoperative colorectal cancer patients should have all 3 tumor markers CEA, CA 19-9, and CA-125 taken within a few days before starting treatment. Tumor markers help preoperatively identify colorectal cancer patients who are at high risk of surgical treatment failure.

Comments on the Quality of English Language

I have no coments

Author Response

Reviewer 2: No comments and no response is required.

Reviewer 3 Report

Comments and Suggestions for Authors

This paper combines a case report and an update/overview on treatment of patients with colon cancer, focusing on patients at high risk for poor oncologic outcome. The manuscript is well written, highly relevant and reports the opinion of the author. 

1. When referring to CT accuracy, section 1.4, the author reports the results of a systematic review and meta analysis by Nerad and quote that sensitivity and specificity was 97% and 81% respectively. This is wrong. The pooled sensitivity and specificity in that study was 90% and 69% respectively, and the sensitivity and specificity reported by the author relates to two CT colonography studies included in Nerads study. This needs to be corrected. In addition, I think it's important to mention that numerous other studies have shown worse diagnostic accuracy of loco regional staging performed by CT to provide a more nuanced overview to the readers.

Local endoscopic resection of early T1 colorectal cancer has increased dramatically the past decades, partly thanks to new endoscopic resection techniques such as ESD as well as national screening programs. I think it´s appropriate to include this in sections mentioning treatment of early colorectal cancer.

Author Response

Reviewer 3:

Comment 1 regarding sensitivity and specificity of CT accuracy reported by Nerad et al.: The pooled sensitivity and specificity was corrected in the manuscript. Also, a paragraph was added to provide the thoughts of a world opinion leader regarding CT assessment of advanced primary colon cancer.

Comment 2 regarding worse diagnostic accuracy of local-regional staging performed by CT:  The reviewer is correct in that CT is a weak tool by which to manage all primary colon cancers. The small early lesions are poorly imaged. CT is not presented to the reader as a screening tool. As the paragraph by Gina Brown establishes, small early colon cancers are poorly characterized, if identified at all, by CT. This manuscript is not about early colon cancer which is adequately treated by minimally invasive colon resection. It is about the poor prognosis primary disease that causes the great majority of deaths when treated in a routine manner by a laparoscopic or robotic resection. Other studies to survey the utility of CT for all primary colon cancers were not added to the manuscript.

Comment 3 regarding T1 colorectal cancer and national screening programs: As stated early in my abstract, this patient presentation and case discussion is about advanced primary colon cancer. A section devoted to treatment of early colon cancer is not true to the purpose of this manuscript. With the Editor’s permission, I am not going to include a completely different topic (early colon cancer) in the manuscript.